

# Patterns of molecular evolution in a parthenogenic terrestrial isopod (*Trichoniscus pusillus*)

Emily Yarbrough[1,2] and Christopher Chandler[1]

[1] Department of Biological Sciences, State University of New York at Oswego, Oswego, NY, United States of America

[2] Department of Biological Sciences, State University of New York at Binghamton, Binghamton, NY, United States of America

## ABSTRACT

The "paradox of sex" refers to the question of why sexual reproduction is maintained in the wild, despite how costly it is compared to asexual reproduction. Because of these costs, one might expect nature to select for asexual reproduction, yet sex seems to be continually selected for. Multiple hypotheses have been proposed to explain this incongruence, including the niche differentiation hypothesis, the Red Queen hypothesis, and accumulation of harmful mutations in asexual species due to inefficient purifying selection. This study focuses on the accumulation of mutations in two terrestrial isopods, *Trichoniscus pusillus*, which has sexual diploid and parthenogenic triploid forms, and *Hyloniscus riparius*, an obligately sexual relative. We surveyed sex ratios of both species in an upstate New York population and obtained RNA-seq data from wild-caught individuals of both species to examine within- and between-species patterns of molecular evolution in protein-coding genes. The sex ratio and RNA-seq data together provide strong evidence that this *T. pusillus* population is entirely asexual and triploid, while the *H. riparius* population is sexual and diploid. Although all the wild *T. pusillus* individuals used for sequencing shared identical genotypes at nearly all SNPs, supporting a clonal origin, heterozygosity and SNP density were much higher in *T. pusillus* than in the sexually reproducing *H. riparius*. This observation suggests this parthenogenic lineage may have arisen via mating between two divergent diploid lineages. Between-species sequence comparisons showed no evidence of ineffective purifying selection in the asexual *T. pusillus* lineage, as measured by the ratio of nonsynonymous to synonymous substitutions (dN/dS ratios). Likewise, there was no difference between *T. pusillus* and *H. riparius* in the ratios of nonsynonymous to synonymous SNPs overall (pN/pS). However, pN/pS ratios in *T. pusillus* were significantly higher when considering only SNPs that may have arisen via recent mutation after the transition to parthenogenesis. Thus, these recent SNPs are consistent with the hypothesis that purifying selection is less effective against new mutations in asexual lineages, but only over long time scales. This system provides a useful model for future studies on the evolutionary tradeoffs between sexual and asexual reproduction in nature.

Corresponding author
Christopher Chandler,
christopher.chandler@oswego.edu

## INTRODUCTION

The "paradox of sex", the question of why sex is maintained in the wild, is a topic of great debate among evolutionary biologists (*Neiman et al., 2018*). Numerous models suggest that, all else being equal, asexual lineages should have strong advantages over sexual ones. For example, a sexually reproducing organism must allocate resources toward locating a mate, courtship, *etc.* (*Otto, 2009*). There are also overall costs associated with the production of males and recombination (*Lehtonen, Jennions & Kokko, 2012*; *Gibson, Delph & Lively, 2017*). Indeed, a few asexual lineages have been highly successful in nature, *e.g.*, undergoing rapid range expansion or diversifying over time (*Fontaneto et al., 2007*; *Gutekunst et al., 2018*). However, sexual reproduction is prevalent in eukaryotic organisms, suggesting it is maintained by selection (*Otto, 2009*; *Speijer, Lukeš & Eliáš, 2015*). This contradiction has been an ongoing topic of debate for years, and multiple hypotheses have been proposed to explain the prevalence of sex. Some of the more famous examples include the niche differentiation hypothesis, the Red Queen hypothesis (*Van Valen, 1973*), and the accumulation of harmful mutations within the genomes of asexual species (*Neiman et al., 2018*). The niche differentiation hypothesis proposes that sexually reproducing species occupy broader ecological niches or have less niche overlap with asexuals, reducing competition between sexuals and asexuals and counteracting the costs of sex (*Neiman et al., 2018*). The Red Queen hypothesis suggests that host-parasite co-evolution selects for sexual recombination (*Agrawal, 2009*). In this situation, it would be more beneficial to be a sexually reproducing organism as negative-frequency dependent selection would favor rare host genotypes (*Lively, 1987*; *Peters & Lively, 1999*; *Agrawal, 2009*).

Sexual organisms also have the benefit of segregation and recombination to facilitate efficient purifying selection, purging deleterious mutations from the population (*Felsenstein, 1974*; *Kondrashov, 1994*; *Hartfield & Keightley, 2012*; *Hollister et al., 2015*; *MacPherson, Scott & Gras, 2021*). Recombination is especially effective at purging such mutations as it not only prevents disadvantageous mutations from becoming established in a population, but it also makes it more likely for favorable mutations to become fixed within a population (*Felsenstein, 1974*). Asexual organisms do not have this benefit, and selection is therefore less effective at removing any deleterious mutations from their genomes (*Keightley & Otto, 2006*). In this instance, one could expect asexual species to suffer from mutational meltdown because, as each generation reproduces, there would be continued accumulation of harmful mutations within the population, potentially leading to population decline and extinction (*Muller, 1964*; *Lynch et al., 1993*). Indeed, extant asexual lineages are often described as having a "twiggy" phylogenetic distribution (*e.g.*, *Vienne, Giraud & Gouyon, 2013*; *Moreira, Fonseca & Rojas, 2021*), consistent with higher extinction rates, though other models may also explain this pattern (*e.g.*, *Janko et al., 2008*; *Schwander & Crespi, 2009*).

On the other hand, empirical studies on the accumulation of deleterious mutations in asexual clades have yielded mixed results. One study found no evidence of ineffective purifying selection in eight asexual hexapod lineages, using the nonsynonymous to
synonymous divergence ratio (dN/dS) as a proxy (*Brandt et al., 2019*). Likewise, genome-wide studies in mites (*Brandt et al., 2017*), ribbon worms (*Ament-Velásquez et al., 2016*), and fishes (*Kočí et al., 2020*), and a mitochondrial study in wasps (*Yan et al., 2022*), also found no evidence of elevated dN/dS ratios in asexual lineages. On the other hand, the absence of this predicted pattern does not necessarily rule out this hypothesis conclusively, and several recent studies have found some patterns consistent with less effective purifying selection in asexual organisms (*Johnson & Howard, 2007*; *Neiman et al., 2010*; *Henry, Schwander & Crespi, 2012*; *Hollister et al., 2015*; *Lovell et al., 2017*; *Sharbrough et al., 2018*; *Bast et al., 2018*; *Maldonado et al., 2022*).

There are several caveats to these findings that may explain these discrepancies. First, many of these studies are based on mitochondrial sequences or a small handful of nuclear genes, rather than genome-wide datasets, making the generality of these findings difficult to infer. Perhaps more importantly, there are a number of different mechanisms by which parthenogenesis can arise, which is especially important given that many extant asexual lineages evolved recently from sexual ancestors (*Neiman, Meirmans & Meirmans, 2009*). The basis of these transitions in reproductive mode can have important consequences on the initial establishment of genetic variation within an asexual lineage as well as on the transmission of those variants from one generation to the next. For instance, some parthenogenic lineages arise through polyploidy, and in the case of allopolyploids, initial levels of heterozygosity may be elevated, depending on the amount of divergence between the parent sexual lineages (*Jaron et al., 2021*). In other cases, transitions to parthenogenesis may be triggered by endosymbionts (*Cordaux, Bouchon & Grève, 2011*; *Verhulst, Pannebakker & Geuverink, 2023*) or other genetic mechanisms, leading to diploid or even haploid parthenogenic lineages (*Weeks, Marec & Breeuwer, 2001*). Subsequently, differences in cell division can lead to either the maintenance or loss of genetic diversity within lineages. For instance, in apomictic species, embryos are produced essentially mitotically, leading to offspring that are clones of their mother, "locking in" the initial asexual genotype with the exception of new mutations. In contrast, in automictic species, offspring genotypes may range from heterozygous clones of the mother to fully homozygous diploids, depending on the exact mechanisms of recombination, chromosome doubling, and meiosis (*Neiman, Sharbel & Schwander, 2014*; *Jaron et al., 2021*).

Understanding transitions to parthenogenesis, and the factors that may promote or constrain these transitions, requires a consideration of the underlying cellular mechanisms. For instance, tests of the efficacy of purifying selection in parthenogenic species may be confounded by initially high heterozygosity in hybrid lineages. Distinguishing novel mutations that occurred after the transition to asexuality from ancestral variants present in the parent lineages, may permit more powerful tests. Similarly, in apomictic species with clonal reproduction and no recombination, novel mutations are unlikely to become homozygous (with the exception of gene conversion events) (*Jaron et al., 2021*), so analyses exploring the evolutionary fate of new mutations must incorporate heterozygous variants to fully capture these processes.

To test the hypothesis that purifying selection is less effective on new mutations in asexuals in nature using a novel system, we examined the terrestrial isopod species

*Trichoniscus pusillus* and *Hyloniscus riparius*. Few studies on the tradeoffs between asexual and sexual reproduction have been conducted in isopods, despite multiple documented occurrences of parthenogenesis in isopods (*Fussey & Sutton, 1981*; *Johnson, 1986*), allowing this system to provide a novel opportunity for study. *T. pusillus* was chosen specifically because it is known to be present in both diploid sexual and triploid asexual populations in Europe (*Christensen, 1983*). *H. riparius* was chosen as a potential obligately sexual outgroup for comparison in the same taxonomic family (*Schultz, 1965*; *Lins, Ho & Lo, 2017*). These species are especially useful for these types of questions because of terrestrial isopods' frequent associations with *Wolbachia* (*Bouchon, Rigaud & Juchault, 1998*; *Cordaux et al., 2012*), which can alter host reproduction in a number of ways. In terrestrial isopods, *Wolbachia* can induce male-to-female sex reversal as well as cytoplasmic incompatibility (*Sicard et al., 2014*; *Becking et al., 2019*). In other organisms, *Wolbachia* can also induce male-killing (*e.g.*, *Hornett et al., 2006*; *Perlmutter et al., 2019*; *Katsuma et al., 2022*; *Arai, Watada & Kageyama, 2024*) and, notably, parthenogenesis (*Arakaki, Miyoshi & Noda, 2001*; *Weeks & Breeuwer, 2001*; *Kremer et al., 2009*), though these effects have never been documented in isopods to our knowledge.

This study had three overall goals: (1) determine the sex ratios of our local *T. pusillus* and *H. riparius* populations in Oswego, New York, to establish whether they are sexual or asexual; (2), use genotype data to examine the cellular mechanisms underlying parthenogenesis in *T. pusillus*; and (3) investigate patterns of molecular evolution using SNP data to test the hypothesis that new mutations may accumulate in asexual species due to inefficient purifying selection. These goals were achieved by surveying local populations and performing RNA-seq to assemble transcriptomes and identify SNPs for each species to examine patterns of nonsynonymous and synonymous divergence and polymorphism.

## METHODS

Portions of this text were previously published as part of a preprint (https://www.biorxiv.org/content/10.1101/2023.01.03.522635v1.full).

### Study species

Both *T. pusillus* and *H. riparius* are native to Europe, likely having been introduced into North America in the last several centuries, as is true of most other common terrestrial isopods in the northeastern United States (*Jass & Klausmeier, 2000*). In Europe, *T. pusillus* is polymorphic, with a sexual, diploid form, as well as a co-occurring asexual triploid form (*Christensen, 1979*; *Christensen, 1983*). Only diploid sexual reproduction has been documented in *H. riparius*, though this does not necessarily rule out parthenogenesis in this species, as we are not aware of any studies specifically assessing reproductive mode in *H. riparius*. Although belonging to different genera, both are members of the family Trichoniscidae, which first appears in the fossil record around the late Eocene/early Oligocene, roughly 35 MYA (*Schmidt, 2008*; *Broly, Deville & Maillet, 2013*), placing an approximate upper limit on the time since these species diverged. Both species are morphologically similar and, at the molecular level, sequence identity between *T. pusillus* and *H. riparius* at the mitochondrial cytochrome oxidase I (COI) marker (∼80% identity)

is only moderately lower than variation within *T. pusillus* (88%–100% identity among specimens). This is similar to inter-specific divergence between other closely related congeners (*e.g.*, *Armadillidium vulgare* and *A. nasatum*, 83% identity) (*Raupach, Rulik & Spelda, 2022*), and therefore supports *H. riparius* as a suitable species for comparison. (Admittedly, however, intraspecific mitochondrial variation in *T. pusillus* and many other terrestrial isopods is unusually high, which may indicate cryptic diversity, or may be related to reproductive endosymbionts, primarily *Wolbachia*; *Hurst & Jiggins, 2005*; *Raupach, Rulik & Spelda, 2022*). Ideally, we would like to compare asexual, triploid *T. pusillus* individuals to their sexual, diploid conspecifics. However, we included *H. riparius* because we had no guarantee of finding any diploid, sexual *T. pusillus* samples in the absence of prior information on our local study population, and sampling/sequencing constraints limited us from performing wider geographic surveys at this time. Unfortunately, there are no morphological or molecular markers that can easily determine ploidy level (and by proxy, reproductive mode) for *T. pusillus*. At the population level, sex ratios can be used as a proxy for the frequency of parthenogenesis (*Fussey, 1984*), but individual reproductive mode can only be inferred by karyotyping, costly high-throughput sequencing (see below), or rearing individual females apart from males, which is challenging due to the difficulty of rearing individuals in isolation (which results in low survival rates in our lab, perhaps because they occur in aggregations in nature) and their slow life cycle (several months to a year to reach maturity). In any case, other studies of molecular evolution in asexual organisms have used inter-generic comparisons with similar levels of divergence between sexual-asexual pairs (*e.g.*, *Brandt et al., 2019*).

For some analyses, an additional outgroup species was needed. Another member of the Trichoniscidae would be ideal, but no other species in this family were available for sequencing, nor are any public transcriptome data currently available from other members of the Trichoniscidae or Synocheta, the larger clade of about 630 species (*Schmidt, 2008*) to which these species belong; thus this study provides the first transcriptomes from both Trichoniscidae and Synocheta. Where necessary, we chose a representative of the Crinocheta, a clade of around 2,750 species which is well supported as sister to Synocheta (*Schmidt, 2008*; *Dimitriou, Taiti & Sfenthourakis, 2019*). Specifically, we chose *Trachelipus rathkei*, also native to Europe but introduced into North America, sampled at the same field site with transcriptome data generated by our lab in prior studies using similar methods (*Becking et al., 2017*).

### Field surveys

All specimens were obtained from Rice Creek Field Station in Oswego, NY, during the months of June, July, and August 2021. Specimens were collected between 8:00 AM and 9:30 AM, and the same areas were sampled a total of four times over three months. To catch specimens, the bases of trees and fallen logs were surveyed and leaf litter was overturned. Homemade mouth aspirators were used to gather specimens from the ground and keep them in a secure container before they were transported to the lab.

Because *T. pusillus* and *H. riparius* are not readily distinguishable by the naked eye, specimens were observed under a stereomicroscope. Each individual was anesthetized

in a 0.01% clove oil solution, as their small size (adults are only a few mm long) and rapid movement make them difficult to handle. Clove oil has been used to anesthetize aquatic isopods in other studies (*Mojaddidi et al., 2018*), and has previously been used to successfully anesthetize larger terrestrial isopods in our lab.

Specimens were identified to species following *Van Name (1936)* and *Shultz (2018)*. Although these two species appear superficially similar, the number of ocelli can be used to distinguish them: each *T. pusillus* eye has three ocelli, while each *H. riparius* eye only has one (*Van Name, 1936*; *Shultz, 2018*). This was visualized under a microscope by turning each specimen on its side to view the individual's ocelli. To determine the sex of each isopod, the shape of the pleon was observed, with females having rounder scale rows and males having more pointed scale rows (*Shultz, 2018*). Females were also identified through the obvious presence of brood pouches, as many were ovigerous during the study period.

After species identification and sexing, each specimen was placed in an individual deli cup with a substrate of moist soil and leaf litter along with thinly sliced carrots. Specimens were kept separate to prevent the spread of any potential pathogens. This also allowed us to obtain full-sibling groups of offspring from females that were gravid at the time of capture.

## RNA extractions

We used RNA-seq to obtain coding sequences and SNPs in each species, as whole-genome sequencing in terrestrial isopods is costly and complex due to their large and highly repetitive genomes (*Chebbi et al., 2019*; *Becking et al., 2019*; *Russell et al., 2021*). In addition to four wild-caught *T. pusillus* females, we included one set of four pooled *T. pusillus* offspring from one of these females, to compare maternal and offspring genotypes. We also included four *H. riparius* specimens, two males and two females. Prior to RNA extraction, the specimens were suspended in RNA Shield (Zymo Research, Irvine, CA, USA) and stored at −20 °C to preserve RNA. An NEB Monarch RNA Extraction Kit (New England BioLabs, Ipswich, MA, USA) was used according to the manufacturer's instructions. RNA yield and quality were assessed using a NanoDrop and Qubit before being sent to the Genomics and Bioinformatics Core Facility at the University at Buffalo for library prep using an NEB stranded RNA library kit and sequencing on an Illumina NovaSeq 6000. As an outgroup, we also assembled a transcriptome from *Trachelipus rathkei*, another terrestrial isopod found at our field site; we used transcriptome data from two wild-caught *T. rathkei* specimens, one male and one female, previously obtained for another study (*Becking et al., 2017*) (SRA Accession Numbers SRR5198726 and SRR5198727). The sequencing reads (all 2 × 100 bp) obtained for this study have been deposited at the NCBI Sequence Read Archive (BioProject Accession PRJNA916870; SRA accession numbers SRR22938942–SRR22938950).

## Transcriptome assembly

Raw sequence reads were trimmed and filtered using the bbduk tool of the bbmap package (*Bushnell, 2024*) with the options ktrim = r, k = 23, mink = 11, hdist = 1, tpe, tbo for adapter trimming.

Following best practices (*Gilbert, 2013*; *Gilbert, 2019*), we assembled transcriptomes using multiple assemblers and subsequently merged and filtered the assemblies to obtain

a final assembly for each species. We first separately assembled the transcriptome of two samples of each species (one male and one female for *T. rathkei* and *H. riparius*; one adult and the set of pooled offspring in *T. pusillus*) with four different assemblers: Trinity v. 2.14.0 (*Grabherr et al., 2011*), TransLiG v. 1.3 (*Liu et al., 2019*), SOAPdenovo-trans v. 1.0.5 (*Xie et al., 2014*) with k = 31, and RNASpades v. 3.15.4 (*Bushmanova et al., 2019*) using k = 31. We then merged the 8 draft assemblies of each species (two biological samples per species times four assemblers per sample). We then filtered the merged set of transcripts for each species using EvidentialGenes v. 3.18 (*Gilbert, 2013*; *Gilbert, 2019*) to remove redundant transcripts. After filtering, EvidentialGenes produces two sets of transcripts for each assembly: "main" transcripts, and a set of "alternate" transcripts, which are predicted to be alternative splicing variants. Next, we removed transcripts/contigs likely originating from contamination, such as bacteria and other symbionts. To accomplish this, we used BLAST+ v. 2.13.0 (*Camacho et al., 2009*) and diamond v. 2.0.15.153 (*Buchfink, Reuter & Drost, 2021*) to look for hits in the NCBI nt and nr databases, respectively, using an *E*-value threshold of $10^{-25}$. Any transcript with a best hit originating from any taxon outside multicellular animals was discarded as a potential contaminant (transcripts with no hits were retained). Because our samples were multiplexed and sequenced together, we also ran CroCo v. 1.1 (*Simion et al., 2018*) to remove transcripts that may represent cross-contamination between species (*e.g.*, due to index hopping). The resulting transcriptome assemblies were assessed using BUSCO v. 5.3.2 (*Manni et al., 2021*) with the arthropoda odb10 gene set.

## SNP calling, pN/pS, and dN/dS analyses

Trimmed sequence reads were aligned to the assembled transcriptomes of the corresponding species using bwa-mem-2-v.2.3.2 (*Vasimuddin et al., 2019*). Sambamba v. 0.8.2 and samtools v. 1.15.1 were used to index and sort the resulting bam files, and filter out discordantly mapped sequence read pairs (*Tarasov et al., 2015*; *Danecek et al., 2021*). Finally, SNPs were called using freebayes v. 1.3.6 (*Garrison & Marth, 2012*) with the specifications of a minimum mapping quality of 60, minimum coverage of 50, and ploidy being 2 for *H. riparius*, which is diploid, and 3 for *T. pusillus*, which our analyses showed is triploid (see below); these parameter values are fairly stringent, to avoid false positive SNPs. At the mapping stage, sequence reads were aligned to the full transcriptome assembly for each species (*i.e.*, predicted main and alternate transcript sequences); however, SNPs were called only in the main transcripts, to avoid pseudo-replication caused by SNPs found in multiple transcript isoforms. Ploidy levels were investigated by counting the number of sequence reads containing the reference allele and alternate allele at each SNP. For diploid samples, the number of reads bearing the reference allele should be roughly equal to the number of reads carrying the alternate allele at most SNPs; on the other hand, at SNPs in triploid samples, the proportion of reads carrying the reference allele should be either 1/3 or 2/3 for any given SNP (*e.g.*, *Ament-Velásquez et al., 2016*). In downstream analyses, we also retained only SNPs at which every sequenced individual had a sequencing depth of 20 or greater, a genotype quality score of 30 or greater, and where the SNP site overall had a quality score of 30 or greater, as estimated by freebayes. When computing SNP frequencies in each species, *i.e.*, number of polymorphic sites/number of total sites sequenced, the total

**Table 1 Sex ratios.** Observed sex ratio data for *T. pusillus* and *H. riparius* specimens caught in Oswego, NY during summer 2021.

| | Female | Male | Total |
|---|---|---|---|
| *T. pusillus* | 170 | 0 | 170 |
| *H. riparius* | 7 | 4 | 11 |

number of sites (in other words, the denominator) included only sites with a sequencing depth of at least 20 in each sequenced individual, because sites with low sequencing depth might appear monomorphic even if there is a true SNP at that location.

We predicted full-length protein sequences in each assembled transcriptome using Codan v. 1.1 (*Nachtigall, Kashiwabara & Durham, 2021*). We then used the predicted coding regions to estimate rates of nonsynonymous and synonymous polymorphism within each species in protein-coding regions (pN, pS, and pN/pS). To identify orthologs across all three species, we used OrthoFinder v. 2.5.4 (*Emms & Kelly, 2019*). To estimate nonsynonymous and synonymous divergence across species (dN, dS, and dN/dS) we aligned predicted amino acid sequences of single-copy orthologs using MUSCLE v. 5.1 (*Edgar, 2004*), back-converted amino acid alignments to nucleotide alignments using pal2nal v. 14.0 (*Suyama, Torrents & Bork, 2006*) and then used the codeml module of PAML v. 4.9 (*Yang, 2007*). We used the free ratios branch model (model = 1) and performed pairwise comparisons (runmode = −2) between *T. pusillus* and *T. rathkei*, and between *H. riparius* and *T. rathkei*. Codon frequencies were used as free parameters (CodonFreq = 3).

## RESULTS

### Field sex ratios

All the captured *T. pusillus* individuals were female ($n = 170$), whereas *H. riparius* had a mixed sex ratio (Table 1) not statistically distinguishable from 1:1 ($p = 0.549$), albeit with a small sample size ($n = 11$). The lack of males in *T. pusillus* suggests this population either reproduces entirely asexually or harbors a sex ratio distorter such as *Wolbachia* (*Cordaux, Bouchon & Grève, 2011*). The even sex ratio in *H. riparius* is consistent with a diploid sexual population.

### Transcriptome assembly, SNPs, and heterozygosity

We obtained transcriptome assemblies for each species, with high BUSCO completeness scores (<6% BUSCOs missing for all assemblies; Table 2). We identified thousands of putative SNPs in each species (Tables 3–4). Genetic similarity was high in *T. pusillus*, as indicated by shared identical genotypes between wild-caught adults and the mother/pooled-offspring pair at nearly all putative biallelic SNPs (Table 3). Because of the complications inherent to inferring SNP genotypes from transcriptome data, some of the putative SNPs may be false positives due to alternative splicing, recently diverged paralogs, or mapping errors; consistent with this, restricting the analysis to SNPs in transcripts identified as complete, single-copy BUSCO markers resulted in slightly higher genetic similarity among samples in *T. pusillus* (Table 3), but did not change any conclusions. In *H. riparius* and *T. rathkei*, on the other hand, a much smaller fraction of SNPs showed identical genotypes

**Table 2 Transcriptome assembly statistics for the three species examined in this study.** "Main transcripts" refers to the primary transcripts predicted by EvidentialGenes; "alternate transcripts" refers to predicted alternative splice variants of the main transcripts; and "all transcripts" refers to the merged set of main and alternate transcripts. Only SNPs in the main transcripts were counted in the last row (*i.e.*, alternate transcripts were excluded), to avoid potential pseudo-replication caused by SNPs found in multiple splice variants of the same transcript.

| | *T. pusillus* | *H. riparius* | *T. rathkei* |
|---|---|---|---|
| Main transcripts | 34,920 (35.2 Mb) | 25,408 (34.7 Mb) | 33,881 (38.8 Mb) |
| Alternate transcripts | 58,041 (64.9 Mb) | 36,684 (63.7 Mb) | 82,444 (114.3 Mb) |
| All transcripts | 92,961 (100.1 Mb) | 62,092 (98.4 Mb) | 116,325 (153.1 Mb) |
| Complete single-copy BUSCOs (all transcripts) | 248 (24.5%) | 243 (24.0%) | 148 (14.6%) |
| Complete duplicated BUSCOs (all transcripts) | 705 (69.6%) | 757 (74.7%) | 844 (83.3%) |
| Fragmented BUSCOs (all transcripts) | 32 (3.2%) | 8 (0.8%) | 10 (1.0%) |
| Missing BUSCOs (all transcripts) | 28 (2.7%) | 5 (0.5%) | 11 (1.1%) |
| Complete single-copy BUSCOs (main transcripts) | 879 (86.8%) | 929 (91.7%) | 935 (92.3%) |
| Complete duplicated BUSCOs (main transcripts) | 37 (3.7%) | 37 (3.7%) | 37 (3.7%) |
| Fragmented BUSCOs (main transcripts) | 40 (3.9%) | 13 (1.3%) | 17 (1.7%) |
| Missing BUSCOs (main transcripts) | 57 (5.6%) | 34 (3.3%) | 24 (2.3%) |
| Total SNPs/biallelic SNPs with genotype data in all samples (number of samples) | 134,485/5,401 (5) | 53,613/12,676 (4) | 32,542/17,684 (2) |

**Table 3 Genetic similarity among individuals, as assessed by the percentage of polymorphic sites at which different sampled individuals have identical genotypes in each species.** In this analysis, SNPs showing evidence of more than two alleles were excluded.

| | *T. pusillus* | *H. riparius* | *T. rathkei* |
|---|---|---|---|
| All biallelic SNPs | Total SNPS: 3,623 All adults identical: 3,399 (93.8%) Mother-offspring identical: 3,309 (91.3%) Two randomly selected adults identical: 3,525 (97.3%) | Total SNPs: 7,857 All adults identical: 504 (6.4%) Two randomly selected adults identical: 4,651 (59.2%) | Total SNPs: 9,977 Both adults identical: 2,859 (28.7%) |
| Biallelic SNPs in single-copy BUSCO transcripts | Total SNPS: 455 All adults identical: 442 (97.1%) Mother-offspring identical: 429 (94.3%) Two randomly selected adults identical: 448 (98.5%) | Total SNPs: 1,051 All adults identical: 23 (2.2%) Two randomly selected adults identical: 622 (59.2%) | Total SNPs: 1,071 Both adults identical: 301 (28.1%) |

among all sequenced samples, regardless of whether all SNPs or only SNPs in single-copy BUSCOs were counted (Table 3). Because the number of sequenced samples differed across species, we also counted the fraction of SNPs showing identical genotypes for a randomly chosen pair of wild-caught adults in each species and observed similar patterns supporting clonality in *T. pusillus* but not *H. riparius* or *T. rathkei* (Table 3).

To assess ploidy levels, the proportion of reads containing the reference allele *vs.* the alternate allele in heterozygous individuals was estimated for all SNPs in each species (Fig. 1). For *T. pusillus* individuals, there are two noticeable peaks in the frequency of reference allele read count ratios at ~0.33 and ~0.67. This observation shows that these

**Table 4 SNP data for heterozygous SNPs within each individual sequenced as part of this study.** These numbers only include SNPs within predicted full-length coding sequences. Covered sites refer to nucleotide sites with sequencing depth of at least 20 in that individual. Overall SNP frequency, nonsynonymous SNP frequency, and synonymous SNP frequency were calculated as the number of total SNPs, nonsynonymous SNPs, and synonymous SNPs divided by the number of covered sites, covered nonsynonymous sites, and covered synonymous sites, respectively.

| Sample | Accession | Covered Nonsyn. sites | Covered Syn. sites | Nonsyn. SNPs | Syn. SNPs | Overall SNP Freq. | Nonsyn. SNP Freq. | Syn. SNP Freq. | pN/pS |
|---|---|---|---|---|---|---|---|---|---|
| *T. rathkei* F | SRR5198726 | 2,530,937 | 688,108 | 1,521 | 2,820 | $1.35 \times 10^{-3}$ | $6.01 \times 10^{-4}$ | $4.10 \times 10^{-3}$ | 0.1466 |
| *T. rathkei* M | SRR5198727 | 1,738,784 | 474,958 | 976 | 2,530 | $1.58 \times 10^{-3}$ | $5.61 \times 10^{-4}$ | $5.33 \times 10^{-3}$ | 0.1054 |
| *H. riparius* F1 | SRR22938945 | 1,724,304 | 466,443 | 832 | 2,904 | $1.71 \times 10^{-3}$ | $4.82 \times 10^{-4}$ | $6.23 \times 10^{-3}$ | 0.0775 |
| *H. riparius* F2 | SRR22938944 | 3,484,506 | 944,256 | 1,662 | 2,546 | $9.50 \times 10^{-4}$ | $4.77 \times 10^{-4}$ | $2.70 \times 10^{-3}$ | 0.1769 |
| *H. riparius* M1 | SRR22938943 | 1,704,014 | 461,491 | 700 | 1,860 | $1.18 \times 10^{-3}$ | $4.11 \times 10^{-4}$ | $4.03 \times 10^{-3}$ | 0.102 |
| *H. riparius* M3 | SRR22938942 | 1,976,952 | 537,642 | 590 | 1,576 | $8.61 \times 10^{-4}$ | $2.99 \times 10^{-4}$ | $2.93 \times 10^{-3}$ | 0.1019 |
| *T. pusillus* 5 | SRR22938950 | 2,548,636 | 700,295 | 4,300 | 13,816 | $5.58 \times 10^{-3}$ | $1.69 \times 10^{-3}$ | $1.97 \times 10^{-2}$ | 0.0855 |
| *T. pusillus* 75 | SRR22938949 | 2,719,388 | 742,972 | 4,250 | 14,107 | $5.30 \times 10^{-3}$ | $1.56 \times 10^{-3}$ | $1.90 \times 10^{-2}$ | 0.0823 |
| *T. pusillus* 170 | SRR22938948 | 3,316,605 | 910,803 | 6,164 | 19,884 | $6.16 \times 10^{-3}$ | $1.86 \times 10^{-3}$ | $2.18 \times 10^{-2}$ | 0.0851 |
| *T. pusillus* 85 | SRR22938947 | 2,358,416 | 645,793 | 4,065 | 13,592 | $5.88 \times 10^{-3}$ | $1.72 \times 10^{-3}$ | $2.10 \times 10^{-2}$ | 0.0819 |
| *T. pusillus* offspring (of *T. pusillus* 85) | SRR22938946 | 334,041 | 90,801 | 248 | 1,290 | $3.62 \times 10^{-3}$ | $7.42 \times 10^{-4}$ | $1.42 \times 10^{-2}$ | 0.0523 |

*T. pusillus* individuals likely have an 'AAB' karyotype, with the 'A' allele at any given SNP being expressed at approximately twice the level of the 'B' allele corresponding to its higher copy number in the nuclear genome. Curiously, the peak at 0.33 is much higher than the peak at 0.67; this indicates that, for any given SNP, some assemblers may be more likely to call the allele with the lower read count as the 'reference' allele. On the other hand, for *H. riparius* and *T. rathkei*, there is only one noticeable peak in the frequency of alleles near the ~0.5 mark, although the peak is slightly below the 0.5 mark in both species, perhaps because of biases in SNP detection or read alignment. Regardless, these patterns are consistent with triploidy in *T. pusillus* and diploidy in *H. riparius*, further supporting the hypothesis that triploid parthenogenesis is the primary reproductive mode in this population of *T. pusillus*.

To investigate heterozygosity levels among species, the proportion of heterozygous nucleotide sites within each individual was estimated for *T. pusillus*, *H. riparius*, and the outgroup *T. rathkei* (Table 4). *T. pusillus* individuals displayed roughly 0.5–0.6% heterozygosity, as compared to the 0.08–0.17% heterozygosity of *H. riparius* and 0.13–0.15% heterozygosity of the *T. rathkei* outgroup. The elevated heterozygosity in triploid *T. pusillus* suggests it may have arisen *via* hybridization between divergent lineages.

We also acknowledge that allelic imbalance (*i.e.*, unequal expression of the two copies of a gene due to cis-regulatory variation or other factors such as paternal or maternal imprinting) can lead to incorrect genotype calls when using RNA-seq data to genotype SNPs. However, this is unlikely to be a major issue in our dataset for multiple reasons. First, the fact that the distribution of allele read count ratios shows clear peaks at 0.33 and 0.66 in *T. pusillus*, and 0.5 in *H. riparius*, suggests that for most genes, each allele is transcribed roughly proportionally to its relative nuclear copy number, and thus allelic imbalance is

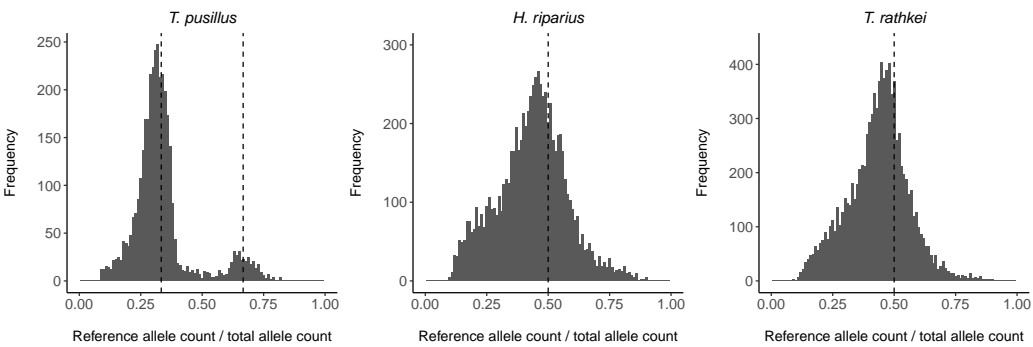

**Figure 1 Allele read count ratio distribution.** Histogram illustrating the distribution of the proportion of reads containing the reference allele (as opposed to the alternate allele) at each SNP in heterozygous individuals within each species. In *T. pusillus*, there are two peaks at approximately 0.33 and 0.66 (vertical dashed lines), consistent with triploid genotypes. In *H. riparius* and *T. rathkei*, there is one peak at approximately 0.5 (vertical dashed lines), consistent with diploid genotypes.

not likely to be a major factor for most genes. Second, a recent study in aquatic isopods identified allelic imbalance in only a small number of genes (<1%), even in "hybrids" between divergent cave and surface lineages (*Lomheim et al., 2023*). Finally, other similar studies have routinely used RNA-seq data for inferring SNP genotypes (*Hollister et al., 2015*; *Bast et al., 2018*; *Brandt et al., 2021*).

### dN/dS and pN/pS

OrthoFinder predicted 11,136 one-to-one (single-copy) orthologs across the three species. We estimated dN/dS ratios using pairwise comparisons, between *T. pusillus* and *T. rathkei*, and between *H. riparius* and *T. rathkei* (Fig. 2). Across these orthologs, *T. pusillus* had significantly higher estimates of dN (0.161 and 0.152 median values for *T. pusillus* and *H. riparius*, respectively; Wilcoxon signed rank test $p = 1.4 \times 10^{-69}$) and dS (7.22 and 6.60 median values for *T. pusillus* and *H. riparius*, respectively; Wilcoxon signed rank test $p = 4.0 \times 10^{-11}$) but significantly lower estimates of dN/dS (median estimates of 0.019 and 0.021 for *T. pusillus* and *H. riparius*, respectively; Wilcoxon signed rank test $p = 1.2 \times 10^{-6}$). However, the differences appear small in magnitude, and the distributions of dN, dS, and dN/dS overlap almost entirely for the two species (Fig. 2). These trends were robust to the choice of codon substitution model (codon frequencies assumed to be equal, CodonFreq = 0; codon frequencies estimated from average nucleotide frequencies, CodonFreq = 1; codon frequencies estimated from average nucleotide frequencies at the three codon positions, CodonFreq = 2), although the exact estimates of dN, dS, and dN/dS varied slightly as expected (Fig. S1).

At heterozygous sites within individuals in *T. pusillus* and *H. riparius*, there was no significant difference between species in pN/pS, the ratio of non-synonymous to synonymous polymorphisms at heterozygous sites in individual samples (Table 4; $p = 0.34$, unpaired Wilcoxon signed rank test), although overall rates of heterozygosity were higher in *T. pusillus* as described above; if anything, pN/pS ratios at heterozygous sites within individuals were slightly lower in *T. pusillus* than in *H. riparius*.

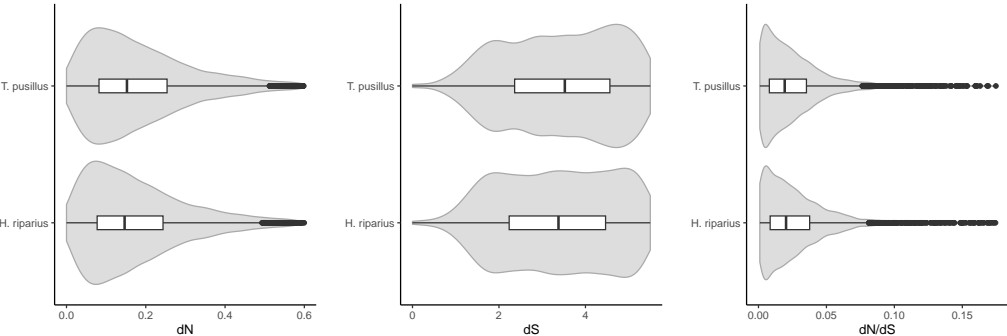

**Figure 2** **dN, dS, and dN/dS.** Violin plots of *dN*, *dS*, and *dN/dS* ratios, resulting from pairwise comparisons between *T. pusillus vs. T. rathkei*, and *H. riparius vs. T. rathkei*. For dN, the *T. pusillus - T. rathkei* median was 0.161; *H. riparius - T. rathkei* median was 0.152; Wilcoxon signed rank test $p = 1.4 \times 10^{-69}$. For dS, the *T. pusillus - T. rathkei* median was 7.22; *H. riparius - T. rathkei* median was 6.60; Wilcoxon signed rank test $p = 4.0 \times 10^{-11}$. For dN/dS, the *T. pusillus - T. rathkei* median was 0.019; *H. riparius - T. rathkei* median was 0.021; Wilcoxon signed rank test $p = 1.2 \times 10^{-6}$.

Finally, we explored the evolutionary fate of new mutations after the transition to parthenogenesis. Specifically, we tested whether a signature of inefficient purifying selection might be most apparent in these "recent" SNPs. Our rationale was that most SNPs within our *T. pusillus* sample might have been the result of a cross between two distinct ancestral lineages, prior to or coinciding with the transition to asexuality; if so, this set of older SNPs would have already been filtered by effective purifying selection in the sexual lineages. This possibility is raised by two observations: first, the density of heterozygous sites in *T. pusillus* individuals is several-fold higher than in either *H. riparius* or *T. rathkei* individuals (Table 4); and second, nearly all SNPs in *T. pusillus* are heterozygous sites within individuals rather than differences between individuals, whereas in *H. riparius* the opposite is true (Table 3). The fact that all of the randomly sampled wild-caught adults share identical genotypes at most SNPs and are just as similar to one another as the parthenogenic mother/pooled-offspring pair suggests that these wild-caught adults descend from a common asexual ancestor. If this is the case, SNPs at which different *T. pusillus* individuals in our sample have different genotypes might represent mutations that occurred after their divergence from this common ancestor. Therefore, we searched for SNPs where wild-caught adult *T. pusillus* individuals (excluding the pooled offspring) had different genotypes, such that some individuals were homozygous for one, putatively ancestral allele, and other individuals carried one copy of a putative derived "mutant" allele. In other words, we filtered for SNPs where some individuals had the genotype 0/0/0 and others had the genotype 0/0/1, with 0 and 1 representing the putative ancestral and derived alleles, respectively. We then looked at the ratio of nonsynonymous to synonymous polymorphisms in this set of candidate recent SNPs (Table 5). The pN/pS ratio at these SNPs, putatively arising from recent mutations, was 0.193, significantly higher than the pN/pS ratio at SNPs in putatively ancestral SNPs *T. pusillus* where all individuals had identical heterozygous genotypes, 0.0577 (likelihood ratio test, null model assuming a

shared pN/pS ratio, alternative model with separate pN/pS ratios for the two sets of SNPs: df $= 1$; $x^2 = 9.02$; $p = 0.0027$). As an additional comparison, we identified SNPs in *H. riparius* with low-frequency alternate alleles (0.125, *i.e.*, only one alternate allele found among the eight alleles sampled in the four sequenced individuals); we might expect these SNPs to also have an elevated pN/pS ratio because alleles with minor deleterious effects should be less likely to reach higher frequencies, and thus we might expect mildly deleterious alleles to be over-represented in this set of SNPs. However, the pN/pS ratio for these low-frequency SNPs in *H. riparius* was 0.0866, still significantly lower than the pN/pS ratio for candidate novel SNPs in *T. pusillus* (likelihood ratio test as above: df $= 1$; $x^2 = 4.1$; $p = 0.042$).

# DISCUSSION

## Apomictic parthenogenesis and triploidy in *T. pusillus*

Multiple lines of evidence support the idea that this population of *T. pusillus* in Oswego, NY, is asexual, with clonal, apomictic reproduction. First, all specimens gathered from Rice Creek Field Station were female, with no males observed out of a sample of 170 individuals. Second, all four wild-caught adults, as well as the offspring of one of these adults, had identical genotypes at nearly all SNPs, consistent with clonal reproduction (Table 3). Indeed, two randomly selected adults show slightly higher levels of similarity by this measure than the mother-offspring pair, which we speculate may be due to higher rates of genotyping errors in the offspring dataset, given that this sample was sequenced at a lower depth and consists of pooled RNA from multiple offspring (which was necessary because individual offspring did not yield sufficient RNA for sequencing). Third, the distribution of the ratio of reference to alternate allele read counts in the sequencing data showed clear peaks at approximately 0.33 and 0.67 (Fig. 1), consistent with triploid genotypes, which are known to be parthenogenic in European populations (*Frankel, 1979*; *Frankel, Sutton & Fussey, 1981*; *Fussey, 1984*). Although *Wolbachia* induces parthenogenesis in other organisms, this effect has not been documented in isopods and is restricted to haplodiploid host species (*Cordaux, Bouchon & Grève, 2011*; *Verhulst, Pannebakker & Geuverink, 2023*), so it seems unlikely that *Wolbachia* is directly responsible for parthenogenesis in this case, although we did not test for *Wolbachia* in our samples. Even so, co-transmission of *Wolbachia* with clonal genomes in asexual lineages may affect dynamics in mixed populations, making this an interesting system for study.

In contrast, *H. riparius* appears to be a typical sexual, diploid, outcrossing population. Although the sample size is limited due to the lower abundance of this species at our field site, the number of males and females sampled is not statistically different from a 50–50 sex ratio. Moreover, SNP data suggest that this *H. riparius* population is diploid (Fig. 1) and outcrossing, with a much smaller proportion of SNPs showing identical genotypes across wild-caught individuals (Table 3).

The observation that all four randomly sampled wild-caught adult *T. pusillus* individuals have identical genotypes at nearly all SNPs (Table 3) suggests that most genetic diversity in this population is due to heterozygosity *within* individuals, rather than sequence differences

Yarbrough and Chandler (2024), *PeerJ*, DOI 10.7717/peerj.17780

**Table 5  SNP data for collections of SNPs across multiple samples in each species.** These numbers only include SNPs within predicted full-length coding sequences, and only sites at which all individuals of that species have sequencing depth of at least 20. "Candidate novel SNPs" refers to SNPs in *T. pusillus* which putatively arose through mutation after the transition to parthenogenesis, in which some individuals had the genotype 0/0/0 and one or more individual had the genotype 0/0/1 . "Identical SNPs" refers to the SNPs in *T. pusillus* which putatively arose prior to the transition to parthenogenesis, in which all individuals sampled had identical heterozygous genotypes (0/0/1). "Low-frequency SNPs" refers to SNPs in *H. riparius* in which the alternate allele had a frequency of 0.125 (*i.e.*, only one heterozygous individual, all others being homozygous for the reference allele), representing a set of SNPs with putatively mildly deleterious alleles for comparison to the *T. pusillus* candidate novel SNPs. Overall SNP frequency, nonsynonymous SNP frequency, and synonymous SNP frequency were calculated as the number of total SNPs, nonsynonymous SNPs, and synonymous SNPs divided by the number of covered sites, covered nonsynonymous sites, and covered synonymous sites, respectively.

| SNP dataset | Covered Nonsyn. sites | Covered Syn. sites | Nonsyn. SNPs | Syn. SNPs | Overall SNP Freq. | Nonsyn. SNP Freq. | Syn. SNP Freq. | pN/pS |
|---|---|---|---|---|---|---|---|---|
| *H. riparius* (sexual)—all SNPs | 881,978 | 237,763 | 657 | 2,289 | $2.63 \times 10^{-3}$ | $7.45 \times 10^{-4}$ | $9.63 \times 10^{-3}$ | 0.0774 |
| *T. pusillus* (asexual) candidate novel SNPs | 1,593,394 | 434,906 | 12 | 17 | $1.43 \times 10^{-5}$ | $7.53 \times 10^{-6}$ | $3.91 \times 10^{-5}$ | 0.1927 |
| *T. pusillus* identical SNPs | 1,593,394 | 434,906 | 338 | 1,602 | $9.57 \times 10^{-4}$ | $2.12 \times 10^{-4}$ | $3.68 \times 10^{-3}$ | 0.0577 |
| *H. riparius* low-frequency SNPs | 881,978 | 237,763 | 388 | 1,209 | $1.43 \times 10^{-3}$ | $4.40 \times 10^{-4}$ | $5.08 \times 10^{-3}$ | 0.0866 |

*between* individuals. The lack of overall genetic differences among individuals suggests that the *T. pusillus* specimens used for this experiment all descend from a common asexual ancestor. Although the number of sequenced individuals is low, if this observation holds up, it would be consistent with either a single, recent origin of parthenogenesis, or a population bottleneck during the establishment of this North American population, likely at some point in the last several centuries (*Jass & Klausmeier, 2000*). Studies on allozyme frequencies in European populations of *T. pusillus* have identified multiple distinct clones in European populations, implying multiple independent origins of parthenogenesis (*Christensen, 1979*; *Friis Theisen, Christensen & Arctander, 1995*), suggesting that the latter explanation for the low diversity among individuals in our population is more likely.

*T. pusillus* displays a much higher density of heterozygous sites within individuals than *H. riparius* (Table 4). These findings provide evidence that this triploid *T. pusillus* lineage may have arisen from a cross between two previously unknown divergent diploid lineages, similar to some other triploid parthenogens (*Moritz, 1983*; *Johnson & Howard, 2007*; *Ament-Velásquez et al., 2016*). Indeed, mitochondrial diversity is high in European populations of *T. pusillus* and other terrestrial isopod species, suggesting the possibility of cryptic diversity, although these patterns may also be driven by co-inheritance with *Wolbachia* (*Hurst & Jiggins, 2005*; *Raupach, Rulik & Spelda, 2022*). An especially interesting parallel is the marbled crayfish, *Procambarus virginalis*, a triploid, parthenogenic lineage first discovered in the aquarium trade in 1995, and now an invasive pest in the wild. Genomic studies have found few differences across its distribution in the wild but much higher heterozygosity in parthenogenic, triploid individuals than in its diploid progenitor, leading to the hypothesis that it arose during a single mating between two distant diploids (*Gutekunst et al., 2018*). Thus, *T. pusillus*, which was likely introduced into North America at least a few centuries ago, as most other common terrestrial isopods in the United States (*Jass & Klausmeier, 2000*), may be an ideal model for studies of how invasive, parthenogenic species diversify and adapt as they spread and become established in new areas.

## Purifying selection in asexual lineages

Patterns of between-species divergence, *i.e.*, dN, dS, and dN/dS, were not consistent with the expectation of inefficient purifying selection in asexual lineages (Fig. 2). Although median dN and dS estimates were significantly higher in the parthenogenic *T. pusillus* than in *H. riparius*, these differences were small in magnitude, and the distributions (among genes in each species) were almost entirely overlapping. Moreover, the median estimated dN/dS ratio was significantly lower in *T. pusillus*, though again, the difference was small, and the distributions of individual dN/dS estimates across genes overlapped substantially. Even if this difference is meaningful, it is in the opposite direction from theoretical predictions; ineffective purifying selection should result in higher dN/dS ratios in the parthenogenic *T. pusillus* lineage. Thus, we find no evidence of inefficient purifying selection in this asexual lineage as predicted by theory (*e.g.*, *Felsenstein, 1974*; *Lynch et al., 1993*; *Kondrashov, 1994*; *MacPherson, Scott & Gras, 2021*) and confirmed in some empirical studies (*e.g.*, *Johnson & Howard, 2007*; *Neiman et al., 2010*; *Henry, Schwander & Crespi, 2012*; *Hollister et al., 2015*; *Bast et al., 2018*; *Maldonado et al., 2022*). However, these results
may be due to the relatively young age of the asexual *T. pusillus* lineage (given that *T. pusillus* occurs in both diploid sexual and triploid asexual forms in its native range; *Frankel, 1979*; *Frankel, Sutton & Fussey, 1981*; *Fussey, 1984*) as there would not have been ample time for nonsynonymous mutations to accumulate. This is especially important because, without recombination or gene conversion, homologous chromosomes will evolve independently, and novel mutations are not expected to become fixed within asexual species, but instead may persist as heterozygous SNPs in what is known as the Meselson effect (*Mark Welch & Meselson, 2000*; *Ament-Velásquez et al., 2016*). Thus, comparisons of assembled transcript sequences may not reveal the full scope of divergence between species if heterozygous transcripts are "collapsed" into a single contig in a transcriptome assembly.

To account for this issue, we also examined pN/pS, the ratio of non-synonymous to synonymous polymorphisms within species, in *T. pusillus* and *H. riparius*. Again, considering all SNPs in each species, we found no significant differences between asexual and sexual lineages (Table 4); if anything, pN/pS ratios are slightly lower in the asexual *T. pusillus*, again counter to theoretical predictions, though this difference is not significant. These results are also not necessarily surprising because, if the asexual *T. pusillus* lineage arose from a cross between two divergent diploid *T. pusillus* lineages, as suggested by its high within-individual heterozygosity, then the majority of the heterozygous SNPs in the asexual lineage would have originated prior to the founding of triploids, as differences between the two divergent diploid lineages, and would have already been filtered by selection in these ancestral sexual lineages.

Finally, we examined the fate of putatively new mutations that occurred after the transition to parthenogenesis, looking at pN/pS ratios considering only the subset of SNPs in *T. pusillus* that differed among individuals (as opposed to putatively ancestral SNPs where all individuals sampled share identical heterozygous genotypes; Table 5). Although the number of such SNPs was small due to our limited sample size, in this subset of putatively recent mutations, pN/pS ratios were significantly higher compared to *T. pusillus* putatively ancestral SNPs where all individuals were identical. One possible alternative explanation for this difference is that these putatively novel or recent SNPs may simply represent a biased sample of mildly deleterious SNPs with low-frequency alternate alleles. However, the pN/pS ratio for this subset of SNPs is also significantly higher than SNPs with comparable low-frequency alternate alleles in *H. riparius* (Table 5). This is consistent with nonsynonymous mutations being purged more quickly in the sexual *H. riparius*, offering some support for the hypothesis that inefficient purifying selection may impose an evolutionary cost on asexual lineages over the long term.

Past studies on the accumulation of deleterious mutations in asexual lineages have yielded mix results, with some studies supporting the hypothesis of inefficient purifying selection (*Johnson & Howard, 2007*; *Neiman et al., 2010*; *Henry, Schwander & Crespi, 2012*; *Hollister et al., 2015*; *Lovell et al., 2017*; *Sharbrough et al., 2018*; *Bast et al., 2018*; *Maldonado et al., 2022*), and others finding limited, conditional, or no evidence (*Ament-Velásquez et al., 2016*; *Brandt et al., 2017*; *Brandt et al., 2019*; *Kočí et al., 2020*; *Yan et al., 2022*). We propose that the inconsistent results in these types of studies may at least partially result from variability in both the biology of the study species and the approaches used. For

instance, evolutionarily older asexual lineages will have had more time for novel mutations to accumulate. Likewise, the mode of parthenogenesis is also expected to impact the rate at which new polymorphisms become homozygous in asexual lineages; for example, in apomictic species, heterozygous SNPs will be maintained across generations (except for presumably rare gene conversion events), whereas in automictic species, heterozygous mutant alleles can become homozygous more readily (*Jaron et al., 2021*). This difference will impact the derived allele's exposure to selection, as well as whether it is detected in, say, a transcriptome-based molecular evolution study. For instance, if a transcriptome assembler randomly selects one allele as the "reference" allele during the assembly process, and only the reference contig sequences are used in dN/dS analyses, up to half of all novel mutations in an asexual lineage may be missed, reducing the statistical power of downstream analyses. And while some studies have considered both within- and between-species sequence variation (*Johnson & Howard, 2007*; *Bast et al., 2018*), this is not universal (*Brandt et al., 2017*; *Brandt et al., 2019*).

There are several limitations to our findings that should be considered. First, ploidy in this system is confounded with reproductive mode (asexuals are triploid, while sexuals are diploid). This is relevant because deleterious mutation accumulation rates may be higher in allopolyploids, even in sexual ones, and may also differ between allopolyploid sub-genomes (*Conover & Wendel, 2022*), so ploidy rather than reproductive mode per se may also be responsible for differences between the diploid sexual and triploid asexual forms. Nevertheless, triploidy is not unusual among asexuals (*Neaves & Baumann, 2011*; *Gokhman & Kuznetsova, 2018*), including some of the most well studied examples (*e.g.*, *Neiman et al., 2010*; *Gibson, Delph & Lively, 2017*), so this system should still provide insights into the evolutionary maintenance of sex in such systems. Moreover, we are unable to distinguish multiple possible explanations for why deleterious mutations may be purged less rapidly in this system—linkage to other polymorphisms caused by the lack of recombination, *versus* "masking" by ancestral alleles due to being "stuck" in a heterozygous state in the absence of gene conversion. However, this is again the case for many systems in which apomictic parthenogenesis has arisen from sexual ancestors, including diploid ones, so this system may still be valuable for understanding the evolutionary maintenance of sex and the evolutionary fate of new mutations in these cases. In addition, comparisons with more closely related outgroups, such as sexual populations of *T. pusillus*, as well as independently derived clones of asexual *T. pusillus*, may provide greater power to detect more subtle differences in patterns of molecular evolution between sexual and asexual lineages. We hope to conduct broader geographic surveys of this species in the future, as well as perform assays for *Wolbachia*, to further tease apart these issues.

## CONCLUSIONS

This study set out to identify the reproductive mode of *T. pusillus* and *H. riparius* populations located in Oswego, NY as well as test the hypothesis that nonsynonymous mutations accumulate more rapidly in asexually reproducing organisms. We confirmed that this population of *T. pusillus* is asexual and triploid, and the sampled population

of *H. riparius* is sexual and diploid. Although patterns of molecular evolution were inconsistent with the hypothesis of inefficient purifying selection in asexual species when considering between-species divergence or all SNPs overall, we did find some initial evidence of an elevated pN/pS ratio for putatively recent mutations. These findings are consistent with less effective purifying selection in asexual lineages only over long time scales, though future work in this system will be useful for confirming this finding. Our results indicate that future studies on the molecular evolution of asexual and sexual lineages should carefully consider both the biology of the study system (*e.g.*, age of the asexual lineage, apomixis *vs.* automixis) as well as methodology for sequence analyses (*e.g.*, accounting for SNPs instead of using only "reference" assembled sequences). We also propose that *T. pusillus*, including studies with wider geographic sampling, should provide a promising novel study system for future research on the evolutionary tradeoffs between sexual and asexual reproduction given its distribution in North America and Europe and the occurrence of both sexual and asexual forms in the wild, particularly in the context of adaptation and diversification during introductions and range expansions and interactions with vertically transmitted endosymbionts.

## ACKNOWLEDGEMENTS

We thank the editor, J. Reinhardt, and anonymous reviewers for helpful comments on earlier versions of this manuscript. We also appreciate computing resources provided by the Biomedical & Health Informatics program at SUNY Oswego and the NSF-supported Jetstream/Jetstream2 cloud computing platforms.

### Funding

This work was supported by the National Science Foundation (NSF DEB 1453298). There was no additional external funding received for this study. The funders had no role in study design, data collection and analysis, decision to publish, or preparation of the manuscript.

### Grant Disclosures

The following grant information was disclosed by the authors:
National Science Foundation: 1453298.

### Competing Interests

The authors declare that they have no competing interests.

### Author Contributions

- Emily Yarbrough conceived and designed the experiments, performed the experiments, analyzed the data, prepared figures and/or tables, authored or reviewed drafts of the article, and approved the final draft.
- Christopher Chandler conceived and designed the experiments, performed the experiments, analyzed the data, prepared figures and/or tables, authored or reviewed drafts of the article, and approved the final draft.

## DNA Deposition

The following information was supplied regarding the deposition of DNA sequences:

The project metadata is available at BioProject: PRJNA916870. The raw sequencing reads are available at SRA: SRR22938942–SRR22938950.

## Data Availability

The code is available in the Supplementary File.

## Supplemental Information

Supplemental information for this article can be found online at http://dx.doi.org/10.7717/peerj.17780#supplemental-information.

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
