# Peer review of "Patterns of molecular evolution in a parthenogenic terrestrial isopod (Trichoniscus pusillus)"

_PeerJ, doi:10.7717/peerj.17780_

## Round 0.1 · original submission · Major Revisions

Dear Drs. Yarbrough and Chandler:

Thanks for submitting your manuscript to PeerJ. I have now received three independent reviews of your work, and as you will see, the reviewers raised some concerns about the research. Despite this, these reviewers are optimistic about your work and the potential impact it will have on research studying mechanisms of evolution in pill bugs. Thus, I encourage you to revise your manuscript, accordingly, considering all the concerns raised by the reviewers.

Thanks for substantially improving the resubmitted manuscript, I enjoyed reading it. Some remarks from the previous reviewers still could be addressed better, such as the fact that you provide ample evidence for the asexuality of T. pusillus versus sexuality in H. riparius. The main improvement here would be to re-cast the research question as something like "what happens to mutations that arise in asexuals?", which would fit with the current goal but raises a few more intuitive subquestions like "how do we identify these mutations?". It seems you are making an ardent attempt to present your data this way but fail to acknowledge that there is an underlying rationale. Please address this in your rebuttal. This likely would require a for more substantial rewrite of the manuscript, whoch may not be a terrible idea considering the other concerns raised by the reviewers.

Many key references seem to be missing in your manuscript; please include these and avoid selecting some papers but not others that serve the same point of reference. If selecting some from many, use “for example”. Some very recent noteworthy studies are called out by the reviewers. Also, please make sure all your references are annotated according to the PeerJ referencing style.

Please ensure that your revision contains accurate accession numbers to your sequences; your data must be made public prior to publication.

Please revise your experimental design for clarity. Your methods should be clearly outlined, and your experiments should be repeatable. All information pertaining to programs, including parameters, should be listed.

There is a lot to work with from these reviews, as all three reviewers have great advice. I am optimistic about your revision and look forward to seeing how you address these concerns and suggestions.

Therefore, I am recommending that you revise your manuscript, accordingly, considering all of the issues raised by the reviewers.

I look forward to seeing your revision, and thanks again for submitting your work to PeerJ.

Good luck with your revision,

-joe

Reviewer 1 ·

Basic reporting

Lines 380-391: It feels like this is the crux of the manuscript; a large chunk is (understandably) devoted to the usual dN/dS ratios and such that we encounter in papers like these, but the real test is in identifying those mutations that have arisen since asexuality has been “triggered” and what their evolutionary fate is. Any pre-existing mutations, while interesting for other reasons, would act to confound the analysis. I think the paper currently does itself injustice by underselling this point. In fact, the preceding section discusses how these mutations can be distinguished from any mutations that are “from a common asexual ancestor” without making it explicit that any mutations that arose afterward are unique in that they represent a direct test of the fixation/persistence rates of mutations that arise in asexual lineages. To me it would make sense to stress this part of the analysis much more to clarify what is in fact the key finding here. This would also help streamline the discussion much more; all other results can be framed as confirming assumptions and whatnot, and this can be cast as the actual testing of the hypothesis.

Lines 312-315: The authors discuss here the peaks in allele reads at 0.33 and 0.67, with the former being much higher. Below, on lines 366-370 the authors discuss a link to a possible hybrid origin of T. pusillus. I think that second bit ought to be considered more up front, e.g. in the introduction. When I first read the passage on 312-315, I figured this might be explained because of hybridization, and this possibility is only mentioned later on. Correct me if I’m wrong, but since these are RNAseq data, aren’t these essentially expression profiles for specific SNPs? If so, the peak at 0.33 being higher means the minor alleles tend to have higher expression; given a possible hybrid origin, I think it makes sense that one ancestral genome (the haploid A, if you will) might have (much) higher expression rates than the other (a diploid B, if you will) in the hybrid T. pusillus (karyotype ABB). There seems to be no linkup from the authors between these two points though.

Experimental design

I think the exclusion of Wolbachia assays in this manuscript is very unfortunate; the authors mention that these are part of their future plans, but I would argue that those data would be valuable here as well. I would urge them to consider performing those assays already so that they can add some more context to their findings; this manuscript could then be referred to in any future work that draws on the results for these assays. Building on this point a linked (albeit minor) point: the authors mention in several places how Wolbachia may contribute to higher (mitochondrial) variation in the host genome, without any explanation about how this effect is brought about. My interpretation would be that Wolbachia diminishes variation, especially for mtDNA, because it spreads through the population alongside with the mtDNA variant with which it is associated. Even in a case with horizontal transmission of Wolbachia, the population-level mtDNA variation would not be increased – or at least, I don’t see how this would generate more mtDNA variation. While I would gladly admit this might be my personal knowledge gap, I can’t help but feel other readers might be similarly confused.

Validity of the findings

No comment, but I feel that the point made under 1. Basic Reporting regarding lines 380-391 could fit here as well as I think this is a key issue regarding how the paper is structured toward the key research question.

Additional comments

Lines 409-412: A recent paper (Verhulst et al. 2023) has proposed that this restriction may be tied to the sex determination cascades in this group, which might be worth citing here as it provides a mechanistic explanation for this restriction. Note: I am not an author on this manuscript.
Verhulst, E.C., Pannebakker, B. & Geuverink, E. (2023). Variation in sex determination mechanisms may constrain parthenogenesis-induction by endosymbionts in haplodiploid systems. Current Opinion in Insect Science 56:101023

Line 464: I would suggest referring also to those original theoretical predictions, i.e. something like this: “Thus, we find no evidence of inefficient purifying selection in this asexual lineage as predicted by theory [REFS] and confirmed in some empirical studies (Johnson and Howard 2007; Neiman et al. 2010; Henry et al. 2012; Hollister et al. 2015; Bast et al. 2018; Maldonado et al. 2022).”

Figure 1: There are some (albeit) minor differences in the X-axis of these figures. I would recommend setting the X-range here from 0 to 1 on all of them. Also (and I realise this sounds really nitpick-y) there is no indication what the dashed lines are supposed to represent. (I mean, I know what the authors are trying to illustrate, but still…)

Figure 2: I would remove the statistical test results from this figure and include them in for example the caption to make the figure less cluttered. The same goes for Supplementary Figure 1. Also, what exactly is the point of doing a statistical test here? The sample sizes are so large that any between-sample differences will be statistically significant, but most likely meaningless from a biological perspective.

DNA data check: The accession number for the BioProject is correct, but the subsequent accession numbers for the actual datasets appear to differ. The described sequencing data sets fit the descriptions elsewhere though, so I think these numbers got updated or I am just looking at the wrong ones myself.

References: The formatting here is pretty inconsistent, but I suppose that’s a job for e.g., a copy editor.

·

Basic reporting

In general this article meets all criteria for basic reporting, it is formatted following PeerJ requirements, properly cited and includes information on obtaining the raw data. Figures and tables use and content is appropriate. I have a few minor comments in this area that should be addressed.

Lines 150 - 153
“reproductive endosymbionts, primarily Wolbachia; Raupach et al. 2022). Ideally, sexual populations of T. pusillus would represent the closest sister taxon to the asexual T. riparius populations for comparisons. However, we included H. riparius because we had no guarantee of finding any diploid, sexual T. riparius samples, and…

This could be written / explained more clearly. I am assuming that you intended to write T. pusillus instead of T. riparius? Or is there also another species in addition to H. riparius and T. pusillus (and T. rathkei)? This seems to be the only part of the paper mentioning this.

Figure 1 & 2
The text in both figures - especially of the X and Y axis - is very small, and was difficult to read on my laptop screen. If this was remedied it would make the data presentation more clear.

Table 2 and Table 3
I think the information on the pages preceding and associated with both figures may have been switched. Table 2 is the assembly statistics and Table 3 is the genetic similarity data, but the text on the pages proceeding implies the opposite.

Table 4
These are information about the sequencing from individuals. However, there does not appear to be a key to what the individual identifiers mean. A column listing the species for each row would be helpful. You could also include the SRR identifier for each for easy access to the correct data.

Experimental design

I have no major critiques in this area. The article is primary research and is in the biological sciences so has the appropriate aim and scope. The research question about patterns of molecular variation in asexual species is well-situated within the literature of the evolution of sex / implications of asexuality. The experiment is designed around the opportunity to study two closely related species of isopods found in New York state in which one is a recent asexual and the other a sexual close relative. The experiment efficiently uses only RNAseq data to address the question about whether purifying selection may be less efficient in asexuals. While using RNAseq only for this type of genomic / molecular evolution analysis may not be the “state-of-the art” approach it once was, it is encouraging to see how much information can be gleaned from a few transcriptomes given the correct analyses are used, as they are here. Furthermore the limitations of this approach (e.g. allele-specific expression) are also clearly and appropriately addressed throughout.

Validity of the findings

I have no concerns about the validity of the findings. The raw data, as mentioned, was submitted to NCBI’s short read archive and I was able to access the data using the provided accession numbers. One interesting and unexpected finding is that the asexual population is Triploid. This means that it is not possible to entirely disentangle the effects of a change in ploidy from the change to asexuality. However, this is discussed and acknowledged repeatedly, and the finding of triploidy itself using the RNAseq data is worth reporting alone. The sample sizes are relatively small for some of the analyses (e.g. the most filtered subset of SNPs in one data set is only 29!), but the conclusions are appropriately measured. The answer to the question about whether purifying selection is in fact weaker in the asexual species is ambiguous, as it depends on the subset of SNPs that are analyzed, and only holds true for the most filtered (and smallest) dataset. However, this is one of those hypotheses that is interesting either way, as a “null” result contradicts theoretical predictions.

Reviewer 3 ·

Basic reporting

I do not have any major concerns with their writing. It's well written, good background and literature citations.

-I like the “Study Species” section and think it is warranted in its current position.
-What sequencing machine/technology was used? Should be added in there around lines 214-223.
-Seems silly to keep “Data cleaning & trimming” as a separate section with one sentence. Could easily be combined with the next section.
-Table 2 and Table 4 need reworking, especially Table 4. Table 2 needs some more details in the legend, as someone who is not intimately familiar with BUSCOs, I had a tough time understanding what all the numbers were. Suggest clarification about that, else it may detract from their findings for other folks that aren’t intimately familiar with this program/process but may have interest.
-Also with Table 2, the authors do not define “all transcripts” in the legend but do define others. Is this the same as “total transcripts”?
-The biggest hang-up for me on this MS is Table 4. I found it very difficult to follow and interpret, it’s just a lot to look at and digest. I think this table needs to be simplified and broken up into multiple tables that explicitly reference numbers/data the authors talk about in text. Most times when the authors referenced this table, it took me a very long time to refresh my memory what the rows and columns all were so I could corroborate their in-text statements (and I found that very difficult to do). As a simple example, on line 326, the authors state a 0.13-0.15% heterozygosity for T. rathkei- where are those numbers in that table? At least I think that’s where the data should be shown, as this is the table the authors reference when talking about heterozygosity. Another, in the discussion (line 435-436) the authors state “T. pusillus displays a much higher density of heterozygous sites within individuals than H. riparius (Table 4).” I cannot make out how Table 4 supports this claim. Between the first column and their legend, I was just constantly confused and couldn’t keep much straight. This table should be reworked and simplified so it’s not a distraction from their points. Focus on the numbers and data they present in-text and build the tables from there- if the authors want to present all these other numbers (which they largely don’t discuss or are confusing) then they could be moved to a supplemental table.
-Line 422: didn’t the authors only randomly compare two individuals, not four? According to Table 3 at least.
-Line 426: “descent” to “descend”
-Lines 427-424: the authors state that “if this observation holds up, it would be consistent with either a single, recent origin of parthenogenesis, or a population bottleneck…” and the authors suggest that a population bottleneck is more likely. Are they mutually exclusive? Couldn’t both be true?
-Lines 466-467: the authors state “However, these results may be due to the relatively young age of the asexual T. pusillus lineage…” where are the data (or citation) that support this lineage as being relatively young? Is this an assumption because they were introduced?
-Lines 483-484: “…as suggest by its high heterozygosity…” suggest clarifying to “…as suggested by its high within individual heterozygosity…” or something similar. Further, “…the asexual linage would have originated earlier…” to ““…the asexual linage would have originated prior to the founding of triploids…”

Experimental design

-I’m slightly confused on their transcriptome assembly/usage. The authors used two samples of each species (so 6 total individuals, 2 per T. pusillus (one was pooled offspring), 2 T. rathkei, and 2 H. riparius) which were “separately assembled” (lines 231-232) but I’m confused which were separately assembled, each species (so two individuals per transcriptome) or all 6 individuals? I think it’s one individual per transcriptome because they then state they “merged the 8 draft assemblies” (that would be 4 programs per individual, so 8 transcriptomes per species; line 236). A bit of clarification would help here.
-With respect to merging the transcriptomes, were there any major differences between the transcriptomes produced by the different assemblers? Or between the two individuals? Any comparison done on them? I imagine if one was way worse, it may not be worth using, and may negatively influence alignment.
-Lines 252-253 the authors state that “Trimmed sequence reads were aligned to the assembled transcriptomes…” More specifics here are very much warranted. Which samples were aligned to which transcriptomes? I assume each species was aligned to their respective transcriptome but did the authors also align multiple species to the same transcriptome? I imagine they must have to estimate dN/dS?
-With respect to variant calling in freebayes, the authors state that a priori they “told” freebayes that T. pusillus was triploid and H. riparius was diploid. Could this bias any SNP/genotype calling? Would any results (especially their allele counts, Figure 1) change if they hadn’t done this a priori to freebayes?
-What about filtering out max depths? With RAD or GBS sequencing, typically we filter out above a certain max depth (e.g., mean+2*SD of depth) to avoid paralogs or gene duplications that can mess with alignment and thus variant calling. Not 100% sure of this issue with RNA seq, especially when one species is triploid, but this is something I want to bring to their attention and perhaps ask why they did not filter above a max depth.
-On lines 295-297 the authors discuss how genetic similarity was high and reference Table 3. I don’t disagree with their findings or statements, but I wonder if these SNPs are real due to their very low apparent frequency. Typically, population genetic studies filter out sites/variants with frequencies less than 5% or 10% frequency (seems most common in the literature) because they are rare and below “detection levels.” Thus, they may not be real and may be due to alignment/calling errors, PCR duplications, or other such things. If anything, this further supports their claims of genetic similarity though since I imagine a lot of them would be filtered out? Again, maybe something the authors could address in text. Other than quality and depth filtering (quality of 30 and depth of 20 was used, lines 263-266) the authors don’t discuss any other filtering steps.
-Do the authors think they lose any resolution by only using biallelic loci in a triploid species? I realize this is a whole new computational ballgame but could be really cool!
-Lines 310-312 (“To assess ploidy levels…) should be added to the methods with a citation where this has been done before.
-Could the different peak heights on Figure 1A be due to not filtering at max depths? What was their average depth across their samples before and after depth filtering? Perhaps a depth of 20 was too high? I imagine filtering could have a large impact on this analysis and should be carefully explored and considered. The authors attribute this to potential assembler bias (lines 314-315), which I buy, but could their use and merging without comparison (at least none is reported) of the different transcriptomes produced by the programs also influence this?
-In Table 3 the mother-offspring identical is lower than two randomly selected adults- which very much seems counterintuitive, ~6% difference between mother and offspring, whereas ~1.5% difference between two randomly selected adults in T. pusillus. Any speculation as to why this occurred and if there is some sort of mechanism introducing genetic variation from mother to offspring?

Validity of the findings

No comment.

Additional comments

no comment.

---

## Round 0.2 · Minor Revisions

Dear Drs. Yarbrough and Chandler:

Thanks for revising your manuscript. The reviewers are generally satisfied with your revision (as am I). Great! However, there are a few concerns to address. Please attend to these issues ASAP so we may move towards acceptance of your work.

Best,

-joe

Reviewer 1 ·

Basic reporting

I appreciate the efforts made by the authors to clarify the underlying goal of identifying mutations that occur after T. pusillus became asexual. Thanks also for clarifying the difference between the peaks at 0.33 vs. 067 and how they come to be.

Minor comments:
Line 148 (or nearby): Maybe add male-killing as a reproductive manipulation? It’d be rather weird to leave one out.

Line 189: “T. pusillus”, not “T. riparius”?

Line 333: Maybe explicitly add “(n=11)” for the sample size?

Experimental design

Regarding the Wolbachia assays: I appreciate the response by the authors and their explanation about why including these in the present study is not feasible. As indicated previously, those assays would have been useful to further support the work presented here, but I feel like the authors make a good point and I think it would be good to be reasonable about this. Hopefully somewhere down the line they will be able to carry out those experiments and revisit some of the thing suggested here.

Validity of the findings

No further comments.

Additional comments

Initial comment: DNA data check: The accession number for the BioProject is correct, but the subsequent accession numbers for the actual datasets appear to differ. The described sequencing data sets fit the descriptions elsewhere though, so I think these numbers got updated or I am just looking at the wrong ones myself.
Response from authors: We are not sure which accession numbers the reviewer is referring to, but the ones given in the text of the manuscript (“BioProject Accession PRJNA916870; SRA Accession Numbers SRR22938942-SRR22938950”) do appear to be correct; we double checked by entering them into NCBI, and those accession numbers took us to the expected samples.
Reviewer response: It seems like the original confusion relates to different IDs assigned to these data sets. The BioProject link has a series of numbers starting with SRX, but when I check these the data sets list a run number that matches those starting with SRR as provided by the authors. The data can be found with those SRR numbers, it’s just that they are not listed under those numbers within the specific BioProject. The SRX numbers range from SRX18896262-SRX18896270, and are found when going to “SRA” in the menu on the right on the BioProject website (https://www.ncbi.nlm.nih.gov/bioproject/PRJNA916870/).

Reviewer 3 ·

Basic reporting

no comment

Experimental design

no comment

Validity of the findings

no comment

Additional comments

I think the authors did a great job addressing the comments and after reading the manuscript, I feel it's good to go.

---

## Round 0.3 · accepted · Accept

Dear Drs. Yarbrough and Chandler:

Thanks for revising your manuscript based on the concerns raised by the reviewers. I now believe that your manuscript is suitable for publication. Congratulations! I look forward to seeing this work in print, and I anticipate it being an important resource for groups studying mechanisms of evolution in pill bugs. Thanks again for choosing PeerJ to publish such important work.

Best,

-joe